# EFFICIENTLY PRE-TRAINING LANGUAGE MODELS WITH MIXTURES OF CLUSTER-ORIENTED, TRAINABILITY-AWARE EXPERTS

## ABSTRACT

Language models (LMs) are pre-trained on large-scale corpora from diverse data sources, encapsulating knowledge across various domains, with their feature spaces often displaying clustering structures. The mixture of experts (MoE) approach is commonly used to scale up model learning capabilities to handle such complexities; however, the fine-grained learning dynamics at the expert level remain largely unexplored. This work analyzes the spatial and temporal characteristics of these clustering structures and examines their impact on the fine-grained trainability of individual experts. Our analysis builds on the singular spectrum of the feature and Jacobian spaces leading to two key observations. First, a few top singular vectors from the feature matrix are sufficient to capture the layer-wise feature cluster patterns. More interestingly, the maximum singular value of the Jacobian matrix reveals conflicts between different feature clusters, and experts exhibit varying levels of trainability, completing their learning asynchronously during training. Inspired by these insights, we proposed Mixture of Cluster-guided, Trainability-aware Experts (MO-CTE), with an efficient routing method to mitigate inter-cluster conflicts to improve expert trainability and a simple yet effective criterion for early stopping low-trainability experts, thus reducing total training costs. We evaluate the proposed MO-CTE across extensive datasets and tasks. Experimental results indicate that MO-CTE accelerates convergence by approximately 37% in test perplexity and 30% in downstream tasks, and improves performance by 3.68% over baselines when consuming similar computation resources.

## 1 INTRODUCTION

It is a prevalent practice to pre-train language models (LMs) on massive-scale, real-world corpora collected from different sources with knowledge across diverse domains (Paeedeh et al., 2024; Wu et al., 2021; Man et al., 2023; Xi et al., 2024). Existing research (Aharoni & Goldberg, 2020) has already shown that such data can lead to the spontaneous emergence of clusters in the feature space. The mixture of experts (MoE) (Fedus et al., 2021) is, therefore, a widely adopted structure that converts dense layers into sparse mixtures of experts to attain better performance in LMs. However, the fine-grained, expert-level learning dynamics, or trainability more precisely, remain a less-investigated research topic (Cai et al., 2024). In this work, we investigate the detailed training behaviors of LMs in the feature and Jacobian space to improve the expert-level trainability related to the spatial-temporal characteristics of the cluster structure.

Without loss of generality, we start with pre-training a Transformer model (Radford et al., 2019) on mixed datasets (McAuley et al., 2015; Komatsuzaki, 2019; Bird et al., 2008) and portray its learning dynamics of the feature space in Figure 1. Observable clustering structures emerge a few steps after the initial step, a phenomenon more pronounced in deep layers with better distinguishable clustering patterns. We discover that the spatial patterns of clusters can be effectively captured in the low-dimensional space fabricated by a few top singular vectors of the feature matrices, allowing performing feature clustering with negligible computational overheads in this space, as is illustrated in Figure 2(a).

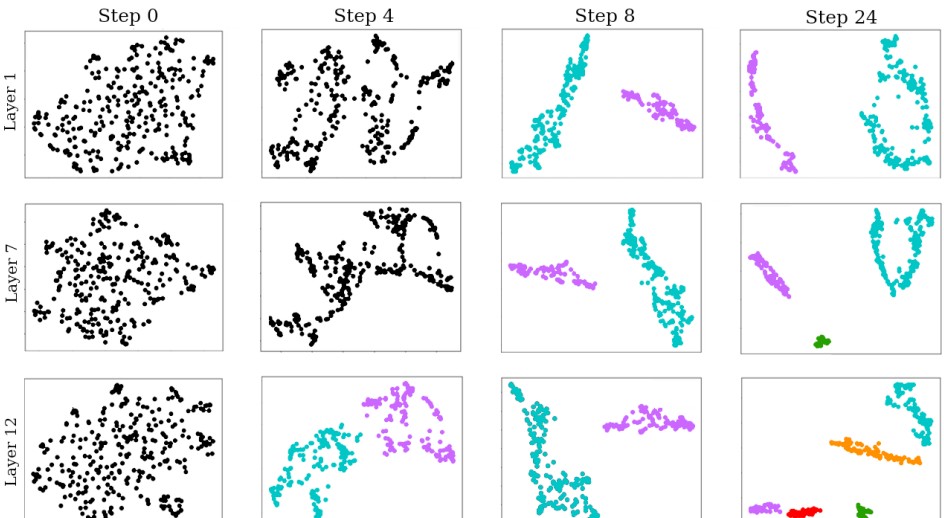

Figure 1: Visualization of feature space in layers 1, 7, and 12 at training steps 0, 4, 8, and 24 respectively. At the start of the training, there is no cluster structure, while features form distinct cluster structures soon. Deep layers show more fine-grained clusters where smaller clusters can be observed.

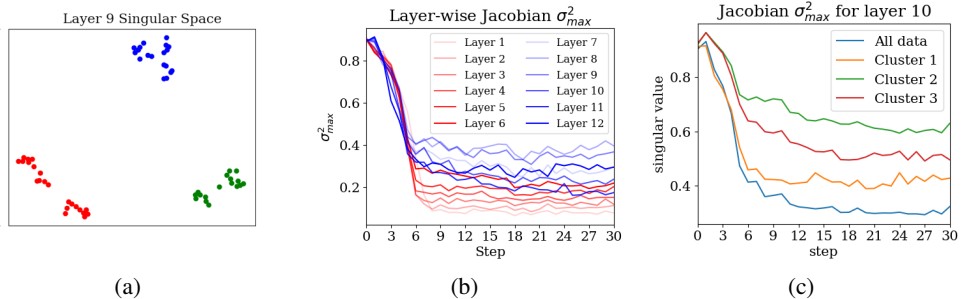

Figure 2: a) A small number of leading singular vectors from feature space are sufficient to fabricate a low-dimensional space maintaining clear cluster structures. b) Evolution of layer-wise Jacobian $\sigma^2_{max}$. c) Variation of Jacobian $\sigma^2_{max}$ of each cluster in layer 10.

Another insight is that the temporal pattern of the maximum singular value of a sample-wise Jacobian matrix, $\sigma^2_{max}$, is synchronized with the changes in cluster structure and the trainability of experts. As is discussed in Jacot et al. (2018b), Fort et al. (2020) and Shi et al. (2024), a high $\sigma^2_{max}$ generally indicates a high level of gradient consistencies of a component, e.g., expert, and thus, its trainability lies in an "informative" space with high learning potentials. Figure 2(b) illustrates the variation of $\sigma^2_{max}$ in all layers, where different experts complete their learning at different times. The cluster structures in shallow layers stabilize soon after they emerge, with a lower $\sigma^2_{max}$ and lower trainability, while deeper layers continue to refine, with a higher $\sigma^2_{max}$ but decreases slowly. Moreover, Figure 2(c) demonstrates that when data from different clusters is mixed during training, these inter-cluster conflicts will lead to poorer overall trainability of experts and inefficient training.

Based on these observations, we propose the Mixture of Cluster-oriented, Trainability-aware Experts (MO-CTE), with an efficient low-dimensional routing method that mitigates inter-cluster feature conflict to improve expert-level trainability and performs early-stopping on experts with low trainability to save computation. Experimental results show that MO-CTE requires about 37% less computation when reaching similar test loss as baseline, and 30% less computation when achieving, and even exceeding the downstream tasks performance as baseline method, with an average improvement up to 3.68% when consuming similar computation resources. Our contributions are summarized as follows:

- We discover that a few top singular vectors in the feature space are sufficient to capture the spatial cluster structures in the feature space.

- We discover the Jacobian $\sigma_{max}^2$ indicates the trainability of different experts, and different experts complete their learning at different times, and mixing cluster data leads to lower trainability of the experts.

- Inspired by the two insights, we propose an MoE variant, namely MO-CTE, with efficient data routing to mitigate inter-cluster conflicts to accelerate intra-cluster learning, and freeze experts at different moments to safeguard expert-level trainability.

## 2 ANALYSIS

In this section, we first formalize the definition of the feature space and feature singular spectrum, to highlight that a few singular vectors of the feature space that interpret the top-k variance are sufficient to capture the different cluster structures. Next, we analyze the Jacobian space and Jacobian singular spectrum, to demonstrate that the largest singular value of Jacobian space, $\sigma_{max}$, can be regarded as an indicator of expert trainability and cluster structure stabilization, while mixed cluster will lead to lower overall trainability.

### 2.1 FEATURE SINGULAR SPECTRUM

The pre-training data for language models are often collected from various sources, containing diverse domain corpora (Man et al., 2023; Paeedeh et al., 2024; Wu et al., 2021), and feature spaces of such data can exhibit cluster structures (Aharoni & Goldberg, 2020). An intuitive example is shown in Figure 1, where in the early stage of training, the cluster structure in the feature space forms quickly, and becomes more pronounced in deeper layers. Visualization of BERT feature space is presented in Appendix B. We aim to describe the characteristics of clusters in the feature space both spatially and temporally, to enhance our understanding of how models learn. We first formally define the feature space and feature singular spectrum.

**Definition 1** *(**Feature Space**) Let $\mathcal{F}(\mathcal{X};\Theta)$ denotes a model $\mathcal{F}$ with its parameters $\Theta$, where $\mathcal{X} \in \mathbb{R}^{n \times d_x}$ denotes training data and $d_x$ is the data dimension. Assume its function can be decomposed into $\{f_1, f_2, ..., f_k, ..., f_L\}$, and parameters can be decomposed into $L$ consecutive exclusive subsets, namely $\Theta = \{\theta_1, \theta_2, \ldots, \theta_L\}$, and $z_{k+1} = f_k(z_k; \theta_k)$. We thus define $z_l \in \mathbb{R}^{n \times d_z}$ as feature spaces where $d_z$ is the model's inner dimension. the model can be represented by $\mathcal{F}(\mathcal{X};\Theta) = f_L(z_L; \theta_L) = f_L(f_{...}(f_2(f_1(x; \theta_1); \theta_2); \theta_{...}; \theta_l)$.*

**Definition 2** *(**Feature Singular Spectrum**) For a feature space $z_l$, we perform Singular Value Decomposition on the feature matrix containing features of data to get its singular values $\{\sigma_{z_l,i}\}_{i=1}^{\min(n,d_z)}$, left singular vectors $\{\mathbf{u}_{z_l,i}\}_{i=1}^{n}$, and right singular vectors $\{\mathbf{v}_{z_l,i}\}_{i=1}^{d_z}$, such that $z_l = \sum_{i=1}^{\min(n,d_z)} \sigma_{z_l,i}\mathbf{u}_{z_l,i}\mathbf{v}_{z_l,i}^{\top}$. We assume all singular values are sorted in descending order, namely $\sigma_{z_l,1} \geq \sigma_{z_l,2} \geq \ldots \sigma_{z_l,k} > 0$. In the following paper, we use singular vectors to refer to the right singular vectors of $z_l$.*

For Transformer-based architectures, we take a layer as the unit of parameter subset. We found that a few leading singular vectors in the Feature Singular Spectrum can form a low-dimensional space where clear cluster structures emerge. Specifically, we select the singular vectors that account for the top 80% of the variance in our study, following the empirical setting in (Abdi & Williams, 2010), and project the feature space onto this low-dimensional space. Representative results from layers 2, 9, and 11 are shown in Figure 3, where the cluster structures are preserved.

This presents an opportunity for efficient cluster-guided MoE learning. Specifically, for different feature clusters, we can introduce the MoE structure and assign features within the same cluster to the same expert. This helps to mitigate inter-cluster feature conflicts and accelerates intra-cluster feature learning, thus improving the trainability of each expert, which will be demonstrated in the next subsection through Jacobian analysis. Recent studies on MoE models also support the idea that their strong performance is due to their ability to assign each cluster of data to a dedicated expert (Chen et al., 2022).

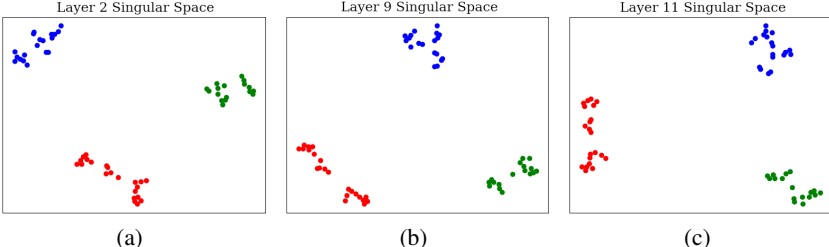

(a)         (b)         (c)

Figure 3: Visualization of the feature space reprojected with the singular vectors corresponding to top-80% variance in a) layer 2, b) layer 9, and c) layer 11. Clear cluster structure is preserved.

Moreover, since we have identified that a low-dimensional space constructed from the top singular vectors of the feature space exhibits clear cluster structures, efficient low-dimensional computations can be sufficient to distinguish different clusters. As a result, we can compute the cluster structures at a negligible low cost and route them to the appropriate experts, further improving the model's learning efficiency and effectiveness.

## 2.2 JACOBIAN SINGULAR SPECTRUM

As shown in the previous section, cluster structures emerge quickly after training begins and stabilize during the mid-to-late stages. This suggests that the model has already captured the cluster features, at which point a strategy is needed to assess parameter trainability and halt training to conserve computational resources.

The Jacobian is defined as the derivative of the model output concerning the model parameters, reflecting the model's learning directions. Following discussions in NTK-related literature (Jacot et al., 2018b; Fort et al., 2020), we adopt the Jacobian matrix and its singular spectrum as indicators of the parameters' trainability. (Khrulkov & Oseledets, 2017). We formally define the Jacobian Matrix and Jacobian Singular Spectrum as follows:

**Definition 3** (*Jacobian Matrix*) *Follow the notations in Definition 1, the j-th row in Jacobian matrix, or $\mathcal{J}_{l,i}$, for a specific parameter group $\theta_l$ is defined as:*

$$\mathcal{J}_{l,j} = \frac{\partial \mathcal{F}(x_j; \Theta)}{\partial \theta_l} \tag{1}$$

*and $\mathcal{J}_l$ is the stack of flattened $\frac{\partial \mathcal{F}(x_j;\Theta)}{\partial \theta_l}$ for all $x_j$ in $\mathcal{X}$.*

**Definition 4** (*Jacobian Singular Spectrum*) *For a Jacobian Matrix $\mathcal{J}_l$, we perform Singular Value Decomposition on it to get its singular values $\{\sigma_{\mathcal{J}_l,i}\}_{i=1}^{\min(n,d_\theta)}$, left singular vectors $\{\mathbf{u}_{\mathcal{J}_l,i}\}_{i=1}^{n}$, and right singular vectors $\{\mathbf{v}_{\mathcal{J}_l,i}\}_{i=1}^{d_\theta}$, such that $\mathcal{J}_l = \sum_{i=1}^{\min(n,d_\theta)} \sigma_{\mathcal{J}_l,i} \mathbf{u}_{\mathcal{J}_l,i} \mathbf{v}_{\mathcal{J}_l,i}^\top$, where $d_\theta$ is the dimension of the parameters. We assume all singular values are sorted in descending order, namely $\sigma_{\mathcal{J}_l,1} \geq \sigma_{\mathcal{J}_l,2} \geq \ldots \sigma_{\mathcal{J}_l,k} > 0$. We normalize the singular values as $\frac{\sigma_{\mathcal{J}_l,i}^2}{\sum_{j=1}^{\min(n,d_\theta)} \sigma_{\mathcal{J}_l,j}^2}$.*

For simplicity and clarity in the subsequent analysis, we will omit the subscript $\mathcal{J}_l$, which denotes the Jacobian matrix of parameter module $l$, and use $\sigma_{max}^2$ to represent the largest normalized singular value. A larger Jacobian $\sigma_{max}^2$ indicates a more dominant learning direction for the parameters across the data, implying they are still in an "informative" space with higher trainability. Conversely, a smaller $\sigma_{max}^2$ suggests that no consistent or dominant learning direction is present, therefore the module is in a "nuisance" space with low trainability.

Figures 4(a) and 4(b) show the variation of the Jacobian $\sigma_{max}^2$ in both GPT and BERT models. According to the figures, the temporal variation of $\sigma_{max}^2$ exhibits a synchronized pattern with the evolution of the cluster structures. Initially, $\sigma_{max}^2$ starts at a relatively high value, indicating that all layers are in an "informative" space with high training potential, and no cluster structures are observed. As training progresses, $\sigma_{max}^2$ drops rapidly, coinciding with the emergence of clusters. In the subsequent stages, the cluster structures in the shallow layers stabilize, and their maximum

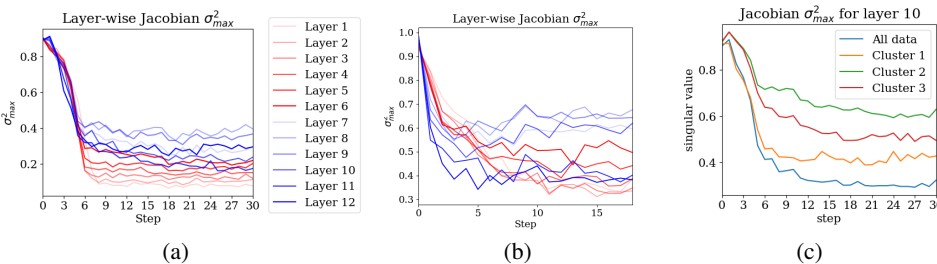

Figure 4: a), b): Evolution of layer-wise $\sigma^2_{max}$ across training steps on a) GPT and b) BERT. Both figures share the same legend. c) Variation of Jacobian $\sigma^2_{max}$ of each cluster in layer 10. We use the cluster label in the late stage to calculate the cluster-wise $\sigma_{max}$ in the early stage.

singular values decrease to a low level and remain constant, suggesting that these layers have entered a "nuisance" space and completed learning the cluster structure, indicating low trainability. However, in deeper layers, the cluster structures continue to refine after their initial emergence, with sub-clusters forming. Their $\sigma^2_{max}$ values show a slow downward trend, indicating that these layers remain in the "informative" space and continue refining and specializing the cluster structure.

Further analysis of cluster-wise $\sigma^2_{max}$ supports our previous assertion that using an MoE approach on cluster-structured feature space can enhance efficiency. Figure 4(c) illustrates the Jacobian $\sigma^2_{max}$ values calculated for each cluster using the parameters of layer 10, where cluster labels from the later stages are applied throughout the entire training process. This result for all layers can be referred to in Appendix C. It is evident that the $\sigma^2_{max}$ values for each cluster drop rapidly, similar to the overall data, but stabilize at a significantly higher level. The cluster structures for these three clusters emerge during this stage but show little further refinement. This suggests that there is still potential for further learning within each cluster to reveal finer sub-cluster structures. However, since these clusters are learned using the same set of parameters, the Jacobians calculated for different clusters conflict with one another, resulting in a smaller overall $\sigma^2_{max}$. Although the parameters remain in the "informative" space, it becomes difficult for them to capture fine-grained cluster features. By assigning different experts to learn different clusters, we can mitigate these conflicts in the Jacobians and allow each expert to focus on the specific characteristics of its assigned cluster. Moreover, our low-dimensional clustering method can improve the efficiency of the MoE by routing data to the appropriate experts with minimal computational cost, thereby enhancing the model's overall efficiency.

## 3 MIXTURE OF CLUSTER-ORIENTED, TRAINABILITY-AWARE EXPERTS

Building on our findings, we propose the Mixture of **C**luster-oriented, **T**rainability-aware **E**xperts (MO-CTE). This approach aims to achieve efficient MoE training with improved expert-level trainability by leveraging the singular spectra of the feature and Jacobian spaces. The MO-CTE strategy consists of two core components: **low-dimensional singularity-based routing** and **safeguarding expert trainability**. With efficient routing for the Mixture of Experts (MoE), changes in trainability determine whether specific experts should be trained or frozen. The real-time behavior of the cluster structure and the Jacobian Singular Spectrum informs these dynamic adjustments. This section details these policies, and the proposed algorithm is outlined in Algorithm 1.

**Low-dimensional Singularity-based Routing** The discussion in Section 2 suggests that the feature space exhibits cluster structures, and features from different clusters may cause conflicts in the Jacobian when learned by the same set of parameters, leading to poorer overall trainability. To address this, we introduce the MoE structure to mitigate inter-cluster feature conflicts and enhance intra-cluster consistency in the features learned by each parameter group, thereby improving the learning efficiency of language models.

Furthermore, with observations in Section 2, we distinguish such cluster structures in a low-dimensional space fabricated by the leading singular vectors in the feature space. With this low-dimensional space, we can achieve efficient feature clustering using very low-dimensional representation to guide the routing for MoE.

In practice, we monitor the cluster structure in this low-dimensional space, taking DBSCAN (Ester et al., 1996) as a clustering algorithm since it does not require an explicit number of clusters as its parameter. Other clustering algorithms like k-Means (MacQueen, 1967) have also been effective in our experiments. The feature matrix is transformed into a low-dimensional space with its singular vectors that interpret 80% variances, and a clustering algorithm is conducted to discover the cluster structure. Suppose the clustering algorithm indicates there exist $k$ clusters. In that case, we introduce the MoE structure with $k$ experts and assign data points with the same cluster to the same expert. To make the learning efficient and optimization landscape smoother, we adopt LoRA-like (Hu et al., 2021) experts with a smaller intermediate dimension and initiate the second matrix with zeros.

---

**Algorithm 1** Mixture of Cluster-oriented, Trainability-aware Experts

---

**Require:** $t$: Interval for monitoring Jacobian.
**Require:** $\alpha, \beta$: Hyper-parameters for judging low trainability.
1: Initialize the $\Sigma^k$ for each expert with its maximal Jacobian Singularity Spectrum $\sigma^2_{max}$.
2: **for** every $t$ steps **do**
3:    **for** each layer in the model **do**
4:       **if** cluster structure is detected in a low-dimensional space **then**
5:          Introduce the MoE structure with a singularity-based routing for each cluster.
6:       **end if**
7:       **if** $\sigma^2_{max,t} < \alpha\Sigma^k$ and $\left|\sigma^2_{max,t} - \sigma^2_{max,t-1}\right| < \beta\Sigma^k$ **then**
8:          Freeze the parameters of a specific expert $k$.
9:       **end if**
10:   **end for**
11: **end for**

---

**Safeguarding Expert Trainability** To ensure only the high trainability experts are updating their parameters to minimize training costs, we apply early-stop according to its Jacobian Singular Spectrum $\sigma^2_{max}$ variation. As discussed in Section 2, a low $\sigma^2_{max}$ indicates a low level of trainability in a "nuisance" space of possible noises and conflicts. Since the $\sigma^2_{max}$ may increase during the early training stage, we record the maximum $\sigma^2_{max}$ of expert $k$ observed in this period as auxiliary criteria, denoted as $\Sigma^k$.

At step $t$ during the following training phase, we continue to monitor the $\sigma^2_{max,t}$ of each given expert $k$ and halt parametric updates when its $\sigma^2_{max,t}$ falls below a specific threshold $\frac{\sigma^2_{max,t}}{\Sigma^k} < \alpha$, and change between $\sigma^2_{max,t}$ and $\sigma^2_{max,t-1}$ falls below a certain threshold relative to $\Sigma^k$: $\frac{\left|\sigma^2_{max,t} - \sigma^2_{max,t-1}\right|}{\Sigma^k} < \beta$.

Where $\alpha$ and $\beta$ are empirically defined hyper-parameters. Through experiments, we found that setting $\alpha$ between 0.10 and 0.20 while $\beta$ between 0.01 and 0.05 typically means a low trainability for an expert, and the cluster structure is also stabilized.

Also, we do not perform Singular Value Decomposition on the Jacobian matrix at every single training step to make the method efficient. Instead, we record the Jacobian Singular Spectrum $\sigma^2_{max}$ over a preset interval. It is proven to work well in our experiments to calculate the cluster structure at every $0.5\% \sim 1.0\%$ training step.

## 4 EXPERIMENTS

In this section, we present the experimental results of applying the Mixture of **C**luster-oriented, **T**rainability-aware **E**xperts (MO-CTE) to models with 140M and 750M parameters. We focus on the GPT architecture (Radford et al., 2019), using data collected from multiple sources for pre-training and evaluating model performance on downstream tasks across various domains. Additionally, we compare the efficacy of MO-CTE with prominent MoE methods, such as Switch Transformers (Fedus et al., 2021).

## 4.1 EXPERIMENTAL SETTINGS

**Datasets** We collected pre-training data from several sources, following general practices in Large Language Model (LLM) training (Zhao et al., 2023). These sources include legal cases (Cas, 2024), medical papers (Cohan et al., 2018), computer science papers (Bird et al., 2008), Amazon reviews (McAuley et al., 2015) and Reddit forums (Komatsuzaki, 2019), simulating a realistic training scenario for LLMs. To evaluate model performance, we also utilize various downstream tasks. A more detailed description of the datasets can be found in Appendix A.

**Evaluation Metrics** Model performance is evaluated based on two key aspects: training efficiency and downstream task performance. For training efficiency, we compute the percentage of floating-point operations (FLOPs) consumed by each model, which serves as a lower bound for execution time (Justus et al., 2018). FLOPs are estimated using the approach described in (Brown et al., 2020) and are reported as a percentage, with the baseline method set to 100%. For downstream performance, we evaluate the models on a range of tasks, using accuracy as the primary metric.

**Calculation of Feature and Jacobian Singular Spectrum** Although defined on the whole dataset, computing the feature and Jacobian singular spectrum on all data can be impractical. So, we estimate the $\sigma_{max}^2$ by randomly sampling a small batch of data and back-propagating on the sum of the logits rather than individual outputs. The batch size is chosen to balance computational feasibility on different hardware, ensuring that the computed $\sigma_{max}^2$ provides a reliable approximation without introducing significant overhead. Also, the $\sigma_{max}^2$ isn't calculated at every single training step. To find a balance between efficiency and evaluation precision, we calculate it at around every 0.8% data.

**Implementation Details** We conduct experiments with GPT-based models at two scales: 140M and 750M parameters (Radford et al., 2019). The 140M model consists of 12 decoder layers with 768 embedding dimensions, 3072 feed-forward network (FFN) dimensions, and 12 attention heads. The 750M model contains 24 decoder layers and 1536 embedding dimensions. Both our method and the Switch Transformer (Fedus et al., 2021) have the same number of experts. The training was performed on NVIDIA GeForce RTX 3090 GPUs for the 140M models and NVIDIA GeForce RTX A100 GPUs for the 750M models, with batch sizes determined by model size and available memory. We used the AdamW optimizer with peak learning rates of $4 \times 10^{-4}$ for the smaller model and $1.5 \times 10^{-4}$ for the larger model. In both models, the intermediate dimension of added LoRA-like experts was set to 1/4 of the original expert. For MO-CTE hyperparameters, we chose $\alpha = 0.20$ and $\beta = 0.05$. More detailed implementation details can be found in Appendix A.

| task | Baseline | MO-CTE(sim. perf.) | MO-CTE(sim. comp.) | Switch |
|---|---|---|---|---|
| test ppl | 130.56 | 120.87 | 90.68 | 107.33 |
| computation | 100.00% | 73.27% | 102.80% | 100.00% |
| CASEHOLD | 48.80 | 50.20 | **50.20** | 50.00 |
| CLIM.SENT. | 62.50 | 66.25 | **68.44** | 66.56 |
| NETZERO | 78.12 | 75.85 | **80.68** | 77.84 |
| SCI-REL | 54.72 | 54.72 | **54.83** | 54.72 |
| RCT-20K | 67.50 | 63.70 | **69.50** | 68.00 |
| SCI-CITE | **75.50** | 71.50 | 75.10 | 70.80 |
| EUADR | 76.92 | **78.63** | 76.64 | 75.50 |
| GAD | 62.80 | **65.20** | 63.00 | 63.70 |
| MRPC | 69.20 | 70.40 | **71.00** | 70.80 |
| QQP | 69.50 | 70.30 | **70.80** | 69.50 |
| Average | 66.56 | 66.68 | **68.02** | 66.74 |

- "sim. perf.": The model employed MO-CTE and achieved a similar performance to the baseline.
- "sim. comp.": The model employed MO-CTE and uses the same computational resources as the baseline.

Table 1: Results for 140M models

## 4.2 RESULTS

**Results of 140M Models** Table 1 presents our experimental results on 140M-scale models trained on the dataset mentioned earlier. Notably, the MO-CTE achieves a comparable test perplexity while using only 73.27% of the computational resources compared to the baseline which reaches a test perplexity of 130.56 with 100% of the computational resources. Both models exhibit similar performance on downstream tasks. When further trained using approximately 100% of the computational resources, MO-CTE surpasses the baseline in downstream task performance. Additionally, MO-CTE outperforms the Switch Transformer with the same number of experts. These results demonstrate that MO-CTE not only enables efficient learning from multi-source heterogeneous data but also leads to improved learning outcomes.

**Results of 750M Models** Table 2 presents the results of the 750M-scale models. For the 750M-parameter model, MO-CTE demonstrates consistent findings with those observed in the 140M model. We achieve comparable downstream task performance while using only about 67.53% of the computational resources, resulting in a reduction of 30% in resource usage. Notably, MO-CTE achieves a comparable test perplexity using just 62.72% of the computational resources compared to the baseline, resulting in a 37% reduction in resource usage. We also recorded the test loss when models used the same computational resources, as shown in Figure 5(a). When fully trained using the same computational resources as the baseline, MO-CTE achieves a lower test perplexity and further improves performance on downstream tasks, with an average improvement of around 3.68%. These experimental results indicate that MO-CTE generalizes well across both model and data scales, demonstrating its effectiveness.

| Task | Baseline | MO-CTE(sim. perf.) | MO-CTE(sim. comp.) | Switch |
|---|---|---|---|---|
| test ppl | 70.10 | 65.17 | 44.81 | 50.21 |
| computation | 100.00% | 67.53% | 100.80% | 100.00% |
| CASEHOLD | 50.20 | 50.00 | **52.50** | 49.80 |
| CLIM.SENT. | 60.62 | 60.31 | **64.06** | 63.13 |
| NETZERO | 84.38 | 84.94 | **87.78** | 85.23 |
| SCI-REL | 54.72 | 54.72 | **59.03** | 58.32 |
| RCT-20K | 68.50 | 69.80 | **72.90** | 72.10 |
| SCI-CITE | 69.10 | 71.80 | **76.80** | 75.30 |
| EUADR | 76.35 | 78.06 | **83.19** | 82.91 |
| GAD | 64.30 | 65.00 | **68.50** | 66.10 |
| MRPC | 71.10 | 71.20 | 71.00 | **71.60** |
| QQP | 72.30 | 72.30 | 72.60 | **72.80** |
| Average | 67.16 | 67.81 | **70.84** | 69.73 |

| Test PPL | Baseline | MO-CTE | Switch | (PFLOPS) |
|---|---|---|---|---|
| 100.00 | 63.71 | 39.82 | 59.16 | |
| 75.00 | 93.30 | 60.68 | 86.64 | |
| 50.00 | 211.63 | 132.74 | 205.93 | |

- "sim. perf.": The model employed MO-CTE and achieved a similar performance compared to the baseline.
- "sim. comp.": The model employed MO-CTE and uses the same computational resources as the baseline.

Table 2: Results for 750M models

**Results of expert-level trainability** We also recorded the proposed Jacobian $\sigma^2_{max}$ metric. After introducing new experts via the expansion strategy, we tracked the changes in each expert's $\sigma^2_{max}$ from the moment of introduction through subsequent training steps, as shown in Figure 5(b). Since experts are introduced at different moments and may be added to different layers, the 0 on the x-axis represents the moment an expert is introduced, rather than implying that all experts are introduced simultaneously. The experiments show that after the introduction of new experts, the Jacobian $\sigma^2_{max}$ for each expert starts at a higher level, indicating better trainability compared to the baseline. Subsequently, it quickly decreases to below the baseline, suggesting that the experts effectively mitigate inter-cluster feature conflict and accelerate intra-cluster feature learning, thus improving the model's overall performance.

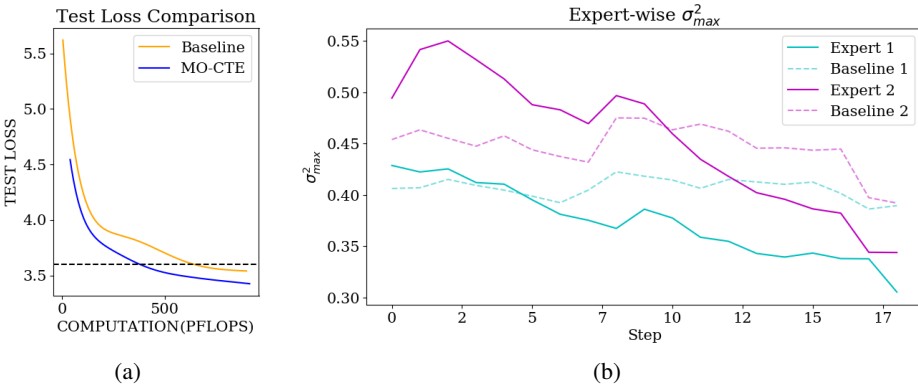

(a)                      (b)

Figure 5: a) We recorded the test loss when models used the same computational resources. The intervention of MO-CTE begins at an early training stage. b) The $\sigma_{max}^2$ of some experts in the model after MO-CTE introduced them, compared with baseline. It shows that the experts show better trainability with each cluster.

## 5 RELATED WORKS

**Mixture of Experts** The Mixture of Experts (MoE) is a key model in machine learning Jacobs et al. (1991); Jordan & Jacobs (1994), where different experts handle distinct regions of the input space. To enhance capacity for complex data, Eigen et al. (2013) extended MoE to deep neural networks, proposing a deep MoE model with multiple layers of routers and experts. Shazeer et al. (2017) improved MoE by making the gating function output sparse, significantly improving training stability and reducing computational cost. Since then, various MoE layers Shazeer et al. (2017); Dauphin et al. (2017); Vaswani et al. (2017) have achieved success in language tasks. MoE has also been used to improve the training efficiency of Large Language Models (LLMs), with routing strategies ranging from token-based selection of experts Lepikhin et al. (2021); Fedus et al. (2022); Zuo et al. (2022); Chi et al. (2022); Dai et al. (2022); Chen et al. (2023), expert-based token selection Zhou et al. (2022), to global expert assignment Lewis et al. (2021); Clark et al. (2022). Inspired by MoE, we propose it can effectively address long-tail knowledge learning.

**Optimization Analysis using Jacobian Spectrum.** Neural Tangent Kernel (NTK) (Jacot et al., 2018a), which calculates the kernel matrix of Jacobian, is known as a powerful tool to analyze convergence and generalization properties (Arora et al., 2019). Many papers (Xiao et al., 2020) study the spectrum of the NTK and find in particular the largest eigenvalue dominates the training regime (Jacot et al., 2018a; Bowman & Montufar, 2022). Multirate training (Vlaar & Leimkuhler, 2022) is a promising technique that partitions neural network parameters into different groups, where the "slow" group is updated less frequently. The mNTK (Shi et al., 2024) further examines fine-grained, module-specific training dynamics and introduces a theoretically motivated method for dynamically adjusting parameter updates based on modular NTK analysis. Additionally, techniques aimed at reducing computational costs, such as network pruning (Lee et al., 2018; Rachwan et al., 2022) and dynamic sparse training (Liu et al., 2020; Jiang et al., 2022), often involve disabling parameters during both forward and backward passes.

## 6 CONCLUSION

In this paper, we studied the layer-wise singular spectrum in both feature space and Jacobian space of language models to achieve a Mixture of Experts (MoE) with improved expert-level trainability. We observed that a few singular vectors in the feature space can capture distinct spatial cluster structures, and the temporal variation pattern of the largest singular value in the Jacobian is synchronized with changes in cluster structure and expert trainability. Based on these observations, we proposed Mixture of Cluster-oriented, Trainability-aware Experts (MO-CTE), which incorporates low-dimensional cluster routing to enhance efficiency and expert-level early stopping to conserve computational resources. Experimental results demonstrate that our approach not only improves the efficiency of MoE learning but also surpasses the performance of baseline methods.

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

## A  EXPERIMENT DETAILS

### A.1  DATASETS

We collected pretrain data from sources of legal (Cas, 2024), medical (Cohan et al., 2018), academical (Bird et al., 2008), web reviews (McAuley et al., 2015) and general fields like Reddit(Openwebtext) (Komatsuzaki, 2019), to simulate the scenario in actual training where data from different sources is gathered and randomly combined into a training dataset. Also, downstream tasks from various domains are used to test the performance of different methods.

Table 3: Datasets used for pretraining

| Pretraining dataset | Description |
| --- | --- |
| Legal( Cas (2024)) | In collaboration with Ravel Law, Harvard Law Library digitized over 40 million U.S. court decisions consisting of 6.7 million cases from the last 360 years into a dataset that is widely accessible to use. |
| PubMed( Cohan et al. (2018)) | PubMed comprises more than 36 million citations for biomedical literature from MEDLINE, life science journals, and online books. |
| Reddit( Komatsuzaki (2019)) | the OpenWebText dataset is an open-source alternative to the WebText dataset, which was used to train OpenAI's GPT models. It consists of web pages curated to exclude content that is difficult to crawl or low-quality, focusing on content similar to that found in Reddit discussions. It is commonly used for training large-scale language models. |
| ACL papers( Bird et al. (2008)) | The ACL Papers dataset contains research papers from the proceedings of the Association for Computational Linguistics (ACL). This dataset provides a wide range of natural language processing (NLP) research papers, including their titles, abstracts, authors, and full-text content. It is useful for tasks such as document classification, citation analysis, and text summarization. |
| Amazon Review( McAuley et al. (2015)) | The Amazon Review dataset consists of millions of product reviews collected from Amazon. The dataset includes information about the reviewer, review text, product ratings, and metadata about the products. It is widely used in research on sentiment analysis, recommendation systems, and opinion mining. |

### A.2  IMPLEMENTATION DETAILS

Table 5 shows the hyperparameters used in our implementations. We use a machine with 8 NVIDIA GeForce RTX 3090 GPUs with 24GB GPU memory and 2 NVIDIA GeForce RTX A100 GPUS with 80GB GPU memory as our experiment platform. Pretraining costs about 30 hours on NVIDIA GeForce RTX 3090 GPUs on and 200 hours on NVIDIA GeForce RTX A100 GPUs.

Table 4: Datasets used for experiments

| Downstream task | Description |
| --- | --- |
| Casehold( Zheng et al. (2021)) | Case Holdings On Legal Decisions, comprising over 53,000+ multiple choice questions to identify the relevant holding of a cited case. |
| GAD( Bravo et al. (2015)) | A relation extraction dataset, to decide if a gene is related to a specific disease. |
| EUADR( van Mulligen et al. (2012)) | Another relation extraction dataset, to decide if a gene is related to a specific disease. |
| Climate Sentiment( Bingler et al. (2023)) | An expert-annotated dataset in environmental fields for classifying climate-related sentiment of climate-related paragraphs in corporate disclosures. |
| Netzero-reduction( Schimanski et al. (2023)) | A dataset for detecting sentences that are either related to emission net zero or reduction targets. |
| QQP( Wang et al. (2019)) | The Quora Question Pairs2 dataset is a collection of question pairs from the community question-answering website Quora. |
| Science-Relation( Beltagy et al. (2019)) | A collection of 500 scientific abstracts annotated with scientific entities, their relations, and coreference clusters. |
| MRPC( Wang et al. (2019)) | The Microsoft Research Paraphrase Corpus (Dolan & Brockett, 2005) is a corpus of sentence pairs automatically extracted from online news sources, with human annotations for whether the sentences in the pair are semantically equivalent. |
| Pubmed-RCT 20k('Dernoncourt & Lee (2017)) | The small 20K version of the Pubmed-RCT dataset by Dernoncourt et al |
| Science Citation( Beltagy et al. (2019)) | A dataset for classifying citation intents in academic papers. |

Table 5: Hyperparameters of Models

| Hyperparameters | 140M GPTs | 750M GPTs |
| --- | --- | --- |
| attention heads | 12 | 16 |
| COP layers | 6 | 24 |
| transformer layers | 12 | 24 |
| Hidden dimension size | 768 | 1536 |
| Droupt | 0.1 | 0.1 |
| Attention dropout | 0.1 | 0.1 |
| Sequence length | 256 | 512 |
| Batch size | 320 | 48 |
| Max steps | 10k | 60k |
| Learning rate decay | Cosine | Cosine |
| $\alpha$ | 0.20 | 0.20 |
| $\beta$ | 0.05 | 0.05 |

## B CLUSTER STRUCTURES IN BERT

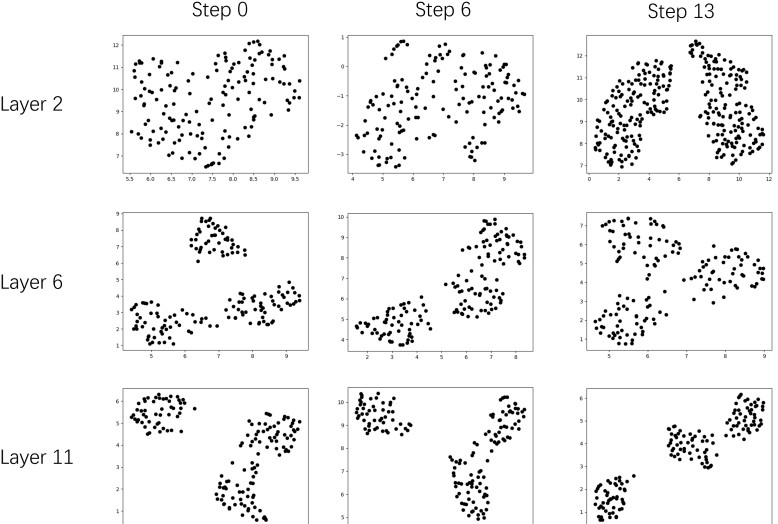

Figure 6: Visualization of feature space in layers 2, 6, and 11 at training steps 0, 6, 13 respectively.

## C  CLUSTE-WISE JACOBIAN $\sigma_{max}$ IN ALL LAYERS

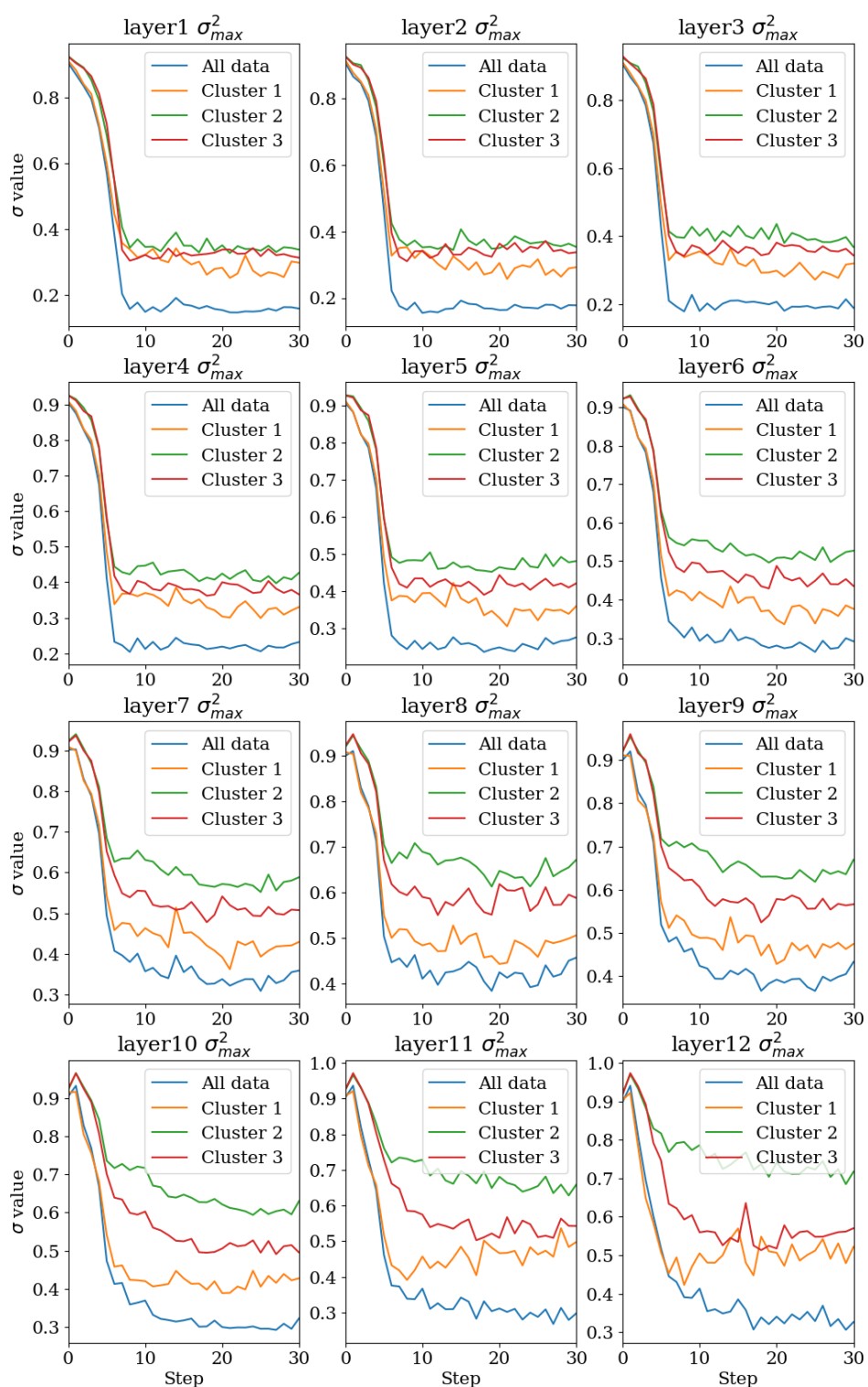

Figure 7: Evolution of Jacobian $\sigma_{max}$ in all layers.

# D COMPLEMENTARY ANALYSIS EXPERIMENT

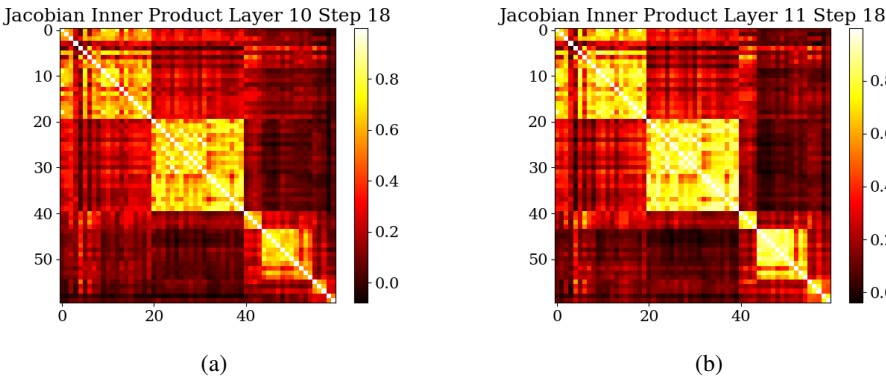

Figure 8: Jacobian cosine similarity(inner product) of all data points in a) layer 10, b) layer 11 at step 18.

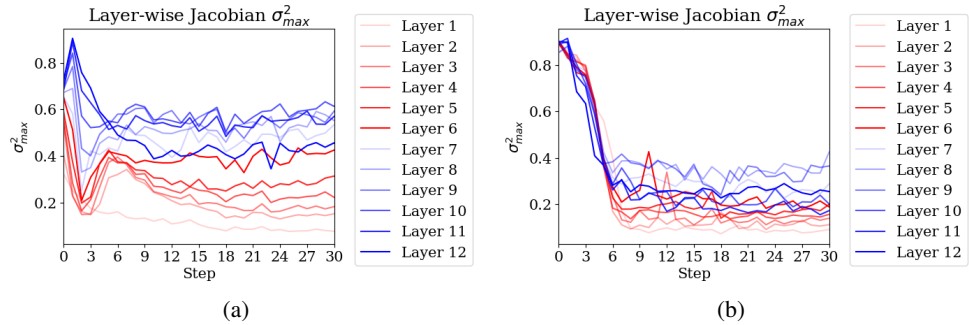

Figure 9: Fine-grained components analysis of $\sigma^2_{max}$ variation in a) attention $W_k$ matrix and b) FFN(MLP).

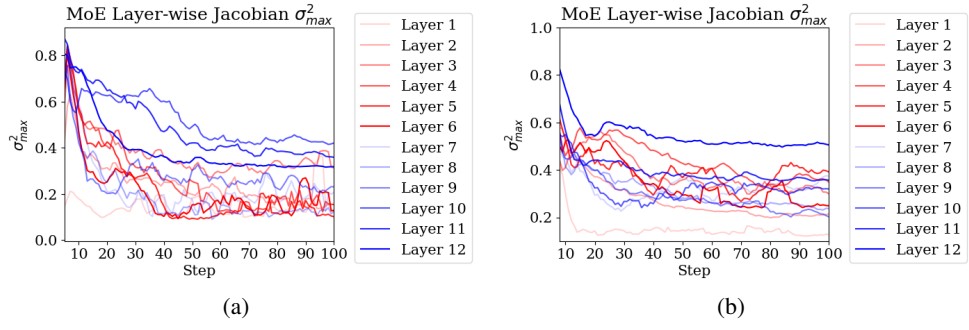

Figure 10: Fine-grained components analysis of $\sigma^2_{max}$ variation in a MoE model, a) attention $W_k$ matrix and b) expert.

# E  DATA SIMILARITY AND JACOBIAN SIMILARITY

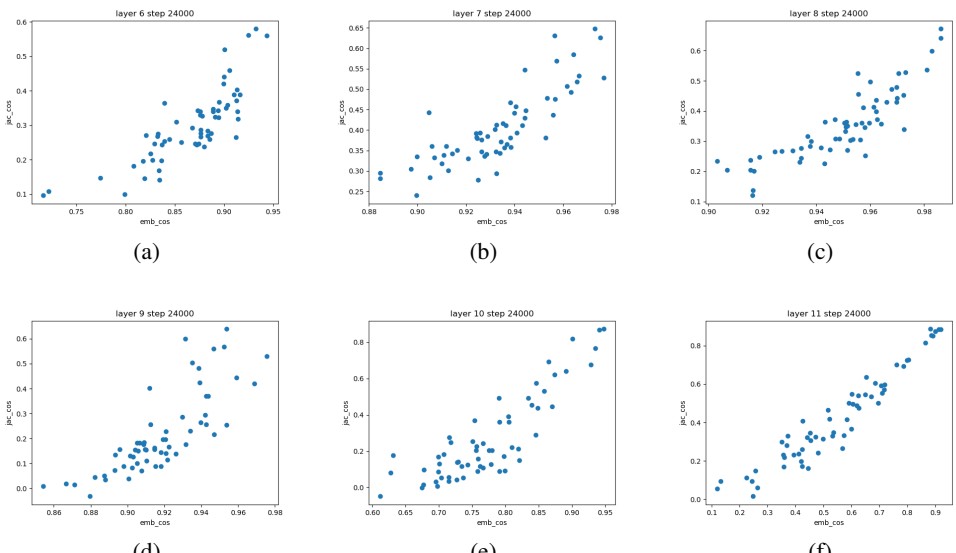

Figure 11: Scatter figures for data's embedding cosine-similarities to their Jacobian cosine-similarities of layers a) 6 to f)11.

