# OpenReview forum: "Efficiently pre-training language models with mixtures of cluster-oriented, trainability-aware experts"
_ICLR.cc/2025/Conference — ICLR 2025 Conference Withdrawn Submission_

### Official Review · Reviewer_jV1G · 2024-10-31

**Soundness:** 3
**Presentation:** 3
**Contribution:** 3
**Rating:** 6
**Confidence:** 3

**Summary:**

The article proposes Mixture of Cluster-oriented, Trainability-aware Experts (MO-CTE), addressing the issue of improving expert-level trainability in language models pre-trained on diverse data sources. It introduces an efficient routing method to mitigate inter-cluster conflicts and early stopping for low-trainability experts. Experiments show MO-CTE accelerates convergence and improves performance with reduced computational resources.

**Strengths:**

1. Overall the paper is well written and easy to read;
2. It focuses on building a trainable MOE at expert-level via exploiting the layer-wise singular spectrum in both feature space and Jacobian space of language models, which is benefit for researchers who encounter the similar problem, and useful for many downstream applications.

**Weaknesses:**

1、In the paper, it seems that many images are identical, such as Figure 2b=Figure 4a, and Figure 2c=Figure 4c. This forces readers to spend time comparing the information described in the images to find any differences. It may also give the impression that this is a tactic to inflate the word count.
2、It is well known that pre-training is a resource-intensive task, requiring a large amount of GPU memory even for small models. However, the author's pre-training setup differs significantly from traditional pre-training settings (e.g., GPT-3). For instance, in Table 5 of the appendix, the author sets the batch sizes for the 140M GPT and 750M GPT models to 40 and 48, respectively, whereas the GPT-3 paper’s Table2.1[1] records a batch size of 500 for models of the same size. It seems that the author is using fine-tuning settings for a pre-training paradigm, raising doubts about the reliability of the conclusions.
3、The dataset tested by the author in the experiments is not a commonly used benchmark, making it difficult to compare with other works. Additionally, the evaluated datasets are primarily classification tasks, lacking evaluation on generation tasks.
4、The author's description of the clustering process is not clear enough. From my understanding, it should involve training for a few steps, then performing clustering once the features are divided into several clusters, and determining the number of experts based on that.
5、What are "LoRA-like experts"? Since LoRA is a fine-tuning architecture, applying it to a pre-training framework would likely result in a low-rank bypass structure. Such an MoE (Mixture of Experts) structure in a pre-training framework would likely struggle to achieve the original SMoE’s goals of expanding dimensions and sparsification.
6、Since the SVD process cannot be accelerated by GPUs, performing SVD at every step is very time-consuming. The inclusion of Algorithm 1 by the author may further slow down the training process. The author needs to provide specific runtime data to demonstrate the algorithm's efficiency.
7. More theory analysis


[1] Language Models are Few-Shot Learners (NeurIPS 2020)

**Questions:**

Please refer to the weakness, thanks.

**Details Of Ethics Concerns:**

NA.

---

### Official Review · Reviewer_J7z2 · 2024-11-01

**Soundness:** 3
**Presentation:** 2
**Contribution:** 2
**Rating:** 3
**Confidence:** 4

**Summary:**

In this paper, the authors aim explore the fine-grained learning dynamics of MoE approaches at the expert level. They got two interesting findings:
1.) a few top singular vectors from the feature matrix are sufficient to capture the layer-wise feature cluster patterns in pre-training LMs;
2.) The maximum singular value of the Jacobian matrix reveals conflicts between different feature clusters, and experts exhibit varying levels of trainability, completing their learning asynchronously during training.
Based on these findings, the authors proposed MO-CTE and empirically demonstrated that MO-CTE can accelerate convergence and improve performance.

**Strengths:**

The authors obtained two interesting findings and proposed a method based on these findings.

**Weaknesses:**

1. The presentation should be significantly improved;
2. Some analysis can not support the authors’ claim;
3. The experiments are not comprehensive and convinced.

**Questions:**

From Figure 1, we see that deep layer (layer 12) shows more fine-grained clusters, Why the authors used layer 9 and layer 10 to demonstrate their claims in Figure 2, demonstrations should be consistent, it’s better to use features from layer 12 in demonstrations shown in Figure 2.

In Line 154, The authors mentioned that “Representative results from layers 2, 9,
and 15 are shown in Figure 3, where the cluster structures are preserved.” While it is obvious that Figure 3 shows features from layers 2, 9, and 11.

In line 211-233, the authors claimed that in Figure 4, the Jacobian values drop rapidly in shallow layers while deep layers continue to refine after the feature clusters’ initial emergence. While Figure 4(b) contradicted with this claim, it is obvious that Layers 11 and 12 drop rapidly than shallow layers. Please provide some explanations.

In Line 244-248, the authors claimed that “By assigning different experts to learn different clusters, we can mitigate these conflicts in the Jacobians and allow each expert to focus on the specific characteristics of its assigned cluster.” While in the experiments, the authors did not provide any extensive experiment to support this claim. Please provide extensive experimental results to support your claim.

In the era of Large language models, lacking experiments on models like llama2-7B is not that convincing.

Only one comparison method made the experimental results less convincing. The authors should include more comparison methods.

---

### Official Review · Reviewer_vTwL · 2024-11-03

**Soundness:** 2
**Presentation:** 2
**Contribution:** 2
**Rating:** 5
**Confidence:** 2

**Summary:**

This paper presents a novel approach for pre-training language models using a method called Mixture of Cluster-oriented, Trainability-aware Experts (MO-CTE). The authors analyze the spatial and temporal characteristics of clustering structures in the feature space of language models and propose an efficient routing method to mitigate inter-cluster conflicts. They also introduce a criterion for early stopping low-trainability experts to reduce training costs. Experimental results demonstrate that MO-CTE accelerates convergence and improves performance compared to baseline methods.

**Strengths:**

1) The paper introduces a novel approach (MO-CTE) that combines cluster-oriented routing with trainability-aware experts.

2) The experimental results are comprehensive and demonstrate significant improvements in convergence speed and performance on downstream tasks.

**Weaknesses:**

1) The paper lacks detailed implementation specifics, such as hyperparameters and training configurations, which are crucial for reproducing the study.

2) There is a need for more extensive evaluation and ablation studies to isolate the contributions of different components of MO-CTE.

**Questions:**

1) Can you provide more detailed implementation details, including hyperparameters and training configurations, to facilitate reproducibility?
﻿
2) Could you conduct additional ablation studies to demonstrate the individual contributions of the cluster-oriented routing and trainability-aware experts in MO-CTE?

---

### Official Review · Reviewer_L4om · 2024-11-04

**Soundness:** 2
**Presentation:** 1
**Contribution:** 2
**Rating:** 3
**Confidence:** 3

**Summary:**

This paper studies how clustering structures in large language models (LLMs)—specifically within the feature space (hidden representations) and Jacobian matrices—affect the training dynamics of Mixture of Experts (MoE) models. Through an analysis, the authors observe that only a handful of dominant singular vectors are sufficient to represent feature cluster patterns at each layer. They also find that the largest singular value of the Jacobian matrix reveals training effectiveness among experts.

Building on these findings, the authors introduce a new approach called Mixtures of Cluster-Guided, Trainability-Aware Experts (MO-CTE). This model incorporates an optimized routing strategy to minimize inter-cluster conflicts and includes an early-stopping criterion for experts with low trainability, effectively reducing overall training costs.

**Strengths:**

* Provides interesting visualizations, such as Figure 1 on clustering structures and Figure 2, which enhance understanding of complex concepts.
* Presents notable observations, such as the significant drop in the Jacobian's maximum singular value during cluster formation (page 4), shedding light on the dynamics of expert learning.

**Weaknesses:**

* The paper assumes that large gradients signify high importance, which may not always be accurate, especially given the heterogeneous scales in the feature space.
* The approach requires substantial computational resources for SVDs and clustering, which are non-trivial. The frequent SVD computations across all layers and heads (every 0.8% of data) are resource-intensive; while random sampling reduces this demand somewhat, it may introduce noise.
* Section 3 could be made clearer with visual illustrations of Algorithm 1 and intuitive examples to enhance reader understanding.
* The model's approach to routing new data points remains unclear, as the paper only addresses data seen during training, which could limit generalizability.
* The absence of downstream evaluations leaves the practical impact of the method on real-world tasks untested.

**Questions:**

* Given that this method relies on SVD on the feature space (data points × feature dimension), can it scale effectively, even with sampling, especially for trillion-token datasets like RedPajama? A discussion on scalability would be beneficial.
* In Tables 2 and 3, does the computation comparison account for the overhead of SVD and clustering etc?
* How are new data points routed efficiently to each expert during test time? This remains unclear.
* Can authors provide downstream evaluations?

---

### Official Review · Reviewer_UNyg · 2024-11-05

**Soundness:** 3
**Presentation:** 3
**Contribution:** 3
**Rating:** 5
**Confidence:** 4

**Summary:**

This paper introduces a novel method, MO-CTE, for pre-training language models via mixtures of cluster-oriented trainability-aware experts. The paper first presents an analysis behind the inherent clustering of a language model's feature space and finds that a small number of the top singular vectors effectively capture the spatial cluster structures within it. The paper then uses these findings to propose the MO-CTE method, which leverages clustering to guide experts to focus on specific clusters and uses the Jacobian to assess expert trainability, which consecutively defines whether a specific expert should be frozen. The proposed method is evaluated on 140M and 750M transformer-based GPT models across a range of metrics, including test perplexity and downstream accuracy on 10 benchmarks.

**Strengths:**

1. The paper's findings in the analysis section are interesting and motivate a novel way to of using SVD and Jacobian analysis to enable more efficient training of mixture-of-expert models.

2. The proposed method is empirically shown to outperform a baseline and the standard "Switch Transformers" mixture-of-experts approach. It is helpful that results are presented by controlling either the accuracy scores or the computation resources.

**Weaknesses:**

1. There are important details that lack clarity in the paper's descriptions, making it hard to understand the findings and methodology.
	1. In the analysis section, the figures lack important details. What is the connection between Figure 2 and Figure 1? Figure 1 shows 5 clusters while figure 2 shows 3. Did the authors verify that the 3 clusters correspond to semantically meaningful topics? What is the data size used to compute these plots and why is the number of bullet points so small?
	2. The experimental setup also lacks details. It is not clear in the main paper which datasets have been used for training, what is their size in tokens and how these datasets were mixed. (While the appendix mentions relevant information there are still details missing and I would be expecting more details in the main paper.)  Additionally, I was not able to find details about the baseline setup.
      3. Also, there are insufficient details in the methodology section, including the exact setup behind the "LoRA-like" experts and the exact criteria behind line 4 of Algorithm 1 ("if cluster structure is detected in a low-dimensional space then"). Moreover, it is hard for the reader to understand how the clusters are initialized at the beginning of the training process, since the algorithm relies on the model's training dynamics.
2. It is unclear how effectively the method can generalize across various settings in practice. The proposed method depends on thresholds ($\alpha$, $\beta$ in Algorithm 1), which might be hard to determine for new models or domains while at the same time there is no quantitative evidence of the method's stability with respect to these thresholds. The analysis is guided by the findings based on the GPT architecture, however it is not clear whether it applies across architectures. For example, the findings do not seem to be as prevalent in the analysis of BERT models in Figure 4(b). Similarly, given the lack of clarity behind the training datasets and its composition, it is unclear whether the findings will apply to realistic LLM training scenarios with large variety of data, including web documents.
3. The paper has limited baselines and misses relevant work. The paper compares to just a single baseline and Switch Transformers while there already exists previous work stemming from the Branch-Train-Merge (Li et al., 2022) literature, including Branch-Train-Mix (Sukhbaatar et al., 2024) and C-BTM (Gururangal et al., 2023). It is unclear, how MO-CTE compares to such work, especially C-BTM which uses a cluster-based approach for routing in mixture-of-experts.


References:
- Li et al., 2022: Branch-Train-Merge: Embarrassingly Parallel Training of Expert Language Models
- Gururangan et al., 2023: Scaling Expert Language Models with Unsupervised Domain Discovery
- Sukhbaatar et al., 2024: Branch-Train-MiX: Mixing Expert LLMs into a Mixture-of-Experts LLM

**Questions:**

- In which model does Figure 4c correspond to?
- How would their observation change if we look at more fine-grained level into transformer architecture (e.g., analyze separately the self-attention and MLP layers)
- Did the authors repeat their analysis on the layers of the trained MoE model?
- How many clusters does DBSCAN return in practice?

---

### Note · Authors · 2024-11-25

I have read and agree with the venue's withdrawal policy on behalf of myself and my co-authors.